# Graph Neural Networks Need Cluster-Normalize-Activate Modules

**Arseny Skryagin**[1]     **Felix Divo**[1]     **Mohammad Amin Ali**[1]
**Devendra Singh Dhami**[2]     **Kristian Kersting**[1,3,4,5]

[1]AI & ML Group, TU Darmstadt     [2]TU Eindhoven     [3]Hessian Center for AI (hessian.AI)
[4]German Research Center for AI (DFKI)     [5]Centre for Cognitive Science, TU Darmstadt

{arseny.skryagin,felix.divo,kersting}@cs.tu-darmstadt.de
amin.ali@stud.tu-darmstadt.de   d.s.dhami@tue.nl

## Abstract

Graph Neural Networks (GNNs) are non-Euclidean deep learning models for graph-structured data. Despite their successful and diverse applications, oversmoothing prohibits deep architectures due to node features converging to a single fixed point. This severely limits their potential to solve complex tasks. To counteract this tendency, we propose a plug-and-play module consisting of three steps: Cluster $\rightarrow$ Normalize $\rightarrow$ Activate (CNA). By applying CNA modules, GNNs search and form super nodes in each layer, which are normalized and activated individually. We demonstrate in node classification and property prediction tasks that CNA significantly improves the accuracy over the state-of-the-art. Particularly, CNA reaches 94.18% and 95.75% accuracy on Cora and CiteSeer, respectively. It further benefits GNNs in regression tasks as well, reducing the mean squared error compared to all baselines. At the same time, GNNs with CNA require substantially fewer learnable parameters than competing architectures.

## 1   Introduction

Graph Neural Networks (GNNs) are a promising approach to leveraging the full extent of the geometric properties of various types of data in many different key domains [Zhou et al., 2020a, Bronstein et al., 2021, Waikhom and Patgiri, 2023]. For instance, they are used to predict the stability of molecules [Wang et al., 2023], aid in drug discovery [Askr et al., 2023], recommend new contacts in social networks [Zhang and Chen, 2018], identify weak points in electrical power grids [Nauck et al., 2022], predict traffic volumes in cities [Jiang and Luo, 2022], and much more [Waikhom and Patgiri, 2023]. To solve such tasks, one typically uses message-passing GNNs, where information from nodes is propagated along outgoing edges to their neighbors, where it is aggregated and then projected by a learned non-linear function. Increasing the expressivity of GNNs is crucial to learning more complex relationships and eventually improving their utility in a plethora of applications.

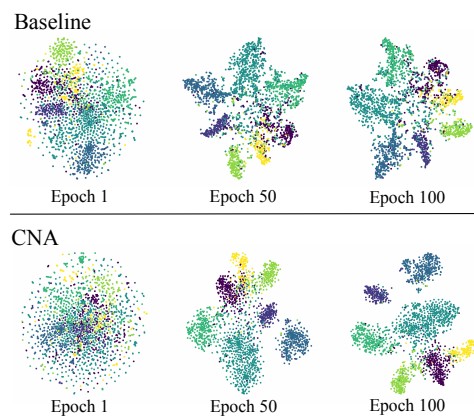

Figure 1: **Evolution of node embeddings for the Cora dataset**. The colors indicate the membership of one of the seven target classes.

38th Conference on Neural Information Processing Systems (NeurIPS 2024).

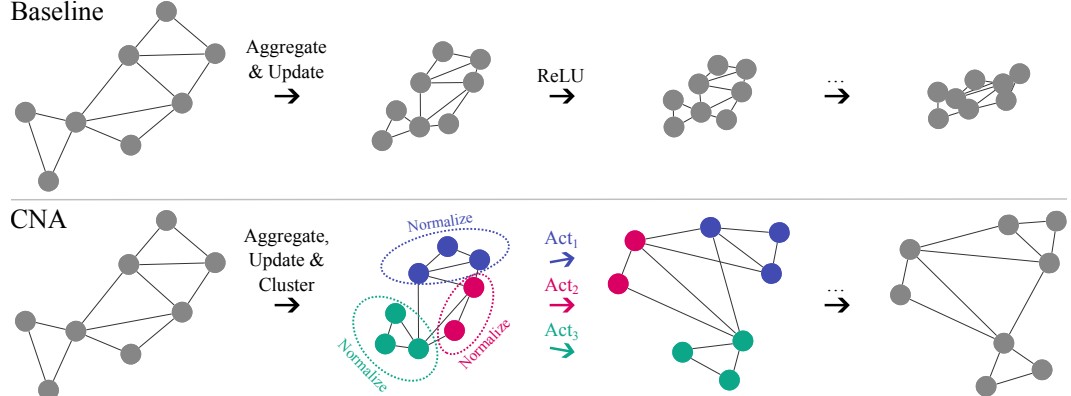

Figure 2: **CNA replaces the activation function in each iteration of any GNN architecture.** When employing classical activations like ReLU to all nodes undifferentiatedly, we observe oversmoothing. With CNA, we cluster the node features and then normalize and project them with a separate learned activation function each, effectively increasing their expressiveness even in deeper networks.

A natural approach to increasing expressivity is to increase depth, effectively enabling further-reaching and higher-level patterns to be captured. This combats under-reaching, where information cannot propagate far enough. For example, this limits the effective radius of information on road crossings in traffic prediction tasks, where information on specific bottlenecks in road networks cannot propagate to the relevant $k$-hop neighbors. In practice, one wants to increase the depth of the employed GNNs. However, this soon triggers a phenomenon called *oversmoothing*, where node features are converging more and more to a common fix-point with an increasing number of layers [NT and Maehara, 2019, Rusch et al., 2023a]. For example, in the specific task of node classification, node features of different classes become increasingly overlapping and, thus, essentially indistinguishable. There are many attempts to prevent this issue from occurring. Among them is Gradient-Gating ($G^2$), which gates updates to nodes once features start converging [Rusch et al., 2023b]. However, $G^2$ adaptively chokes message passing in each node right before oversmoothing can occur, effectively reducing the functionality of deeper GNN layers to an identity mapping. This idea of adaptively controlling the flow of information in each node is still a very promising approach. But, instead of regulating message passing, we propose learning an adaptive node feature update. We argue that it is crucial to ensure that while the node features are iteratively exchanged, aggregated, and projected, they stay sufficiently different from each other to solve the eventual task, like classification or regression. This has the benefit of maintaining effective information propagation even in deeper layers. Figure 1 visualizes the final node features during training, showing how our method improves the separation of the learned classes over the oversmoothed baseline.

To ensure sufficiently distinct nodes, we present Cluster → Normalize → Activate (CNA) modules,[1] specifically designed to improve the expressivity of GNNs:

- **Cluster** – Transformation of the node features should be shared and yet differ at the same time. For this reason, our first inductive bias is to assume several groups of nodes with shared properties.

- **Normalize** – Stabilization of training in deep architectures, including Transformers [Vaswani et al., 2017], is typically provided by normalization. By employing normalization, CNA effectively maintains beneficial numerical ranges and combats collapse tendencies.

- **Activate** – To preserve distinct representations, the clusters must be transformed individually. By introducing learnable activation functions, we learn separate projections for each of them. This generalizes the typical affine transformation following the normalization to a general learned function that can better adjust to the specific node features.

We use rational Activations [Molina et al., 2019, Delfosse et al., 2024] as powerful yet efficient point-wise non-linearities. The complete procedure is shown in Figure 2.

---

[1]Code available at `https://github.com/ml-research/cna_modules`.

CNA modules can also be viewed as adding additional hierarchical structure to the problem: By grouping nodes into clusters of similar representations, we effectively introduce super-nodes with different non-linear activation functions. Each of their constituents shares the same activation function yet has distinct node property vectors and neighbors. Moreover, the node features in each super-node are less varied since the members of the clusters share some common characteristics. This divide-and-conquer approach breaks up the challenging task of transforming the node features into many smaller ones.

The presented work introduces the novel CNA modules which limit oversmoothing and thereby improve performance. They allow for many advancements, delivering better performance compared to the state-of-the-art in many tasks and datasets. In summary, we make the following contributions:

(i) We introduce the plug-and-play CNA modules for more expressive GNNs and motivate their construction.

(ii) We show that they empirically allow training much deeper GNNs.

(iii) Our experiments demonstrate the effectiveness of CNA in diverse node and graph-level classification, node-property prediction, and regression tasks.

(iv) Lastly, we show that architectures with CNA are parsimonious, achieving better performance than the state-of-the-art with fewer parameters.

We proceed as follows: We next relate our work to the existing research on GNNs and their specific challenges (Section 2). We then describe and discuss our proposed solution CNA (Section 3) and conduct a comprehensive evaluation in different scenarios (Section 4). Finally, we conclude and suggest promising next steps for further improving the expressiveness of GNNs (Section 5).

## 2 Related Work

**Machine Learning on Graphs and its Challenges.** Machine learning on graphs has a long history, with graph neural networks as their more recent incarnations [Gori et al., 2005, Scarselli et al., 2008]. Since then, several new models like Graph Convolutional Networks (GCN) [Kipf and Welling, 2016], Graph Attention Networks (GAT) [Veličković et al., 2018], and GraphSAGE [Hamilton et al., 2017] have been proposed. Gilmer et al. [2017] then unified them into the Message Passing Neural Networks (MPNNs) framework, the most common type of GNNs [Battaglia et al., 2018]. In addition to the typical machine learning pitfalls like overfitting and computationally demanding hyperparameter optimization, MPNNs pose some specific challenges: *oversquasching* is the effect of bottlenecks in the graph's topology, limiting the amount of information that can pass through specific nodes [Alon and Yahav, 2020, Topping et al., 2021]. The other widely studied challenge is *oversmoothing*, where the node features converge to a common fixed point with increasing depth of the MPNN [Li et al., 2018, NT and Maehara, 2019, Rusch et al., 2023a]. This essentially equates to the layers performing low-pass filtering, which is harmful to solving the problem beyond some point. This phenomenon has also been studied in the context of Transformers [Vaswani et al., 2017], where repeated self-attention acts similarly to an MPNN on a fully connected graph [Shi et al., 2021a]. Different metrics have since been proposed to measure oversmoothing: cosine similarity, Dirichlet energy, and mean average distance (MAD). Rusch et al. [2023a] organize the existing mitigation approaches into three main groups. First, as discussed in more detail in the next paragraph, normalization and regularization are beneficial and are also performed by our CNA modules. Second, one can change the propagation dynamics, as done by GraphCON [McCallum et al., 2000], Gradient Gating [Rusch et al., 2023b], and RevGNN [Li et al., 2021]. Finally, residual connections can alleviate some of the effects but cannot entirely overcome them. Solving these challenges is an open task in machine learning on graphs.

**Normalization in Deep Learning.** In almost all deep learning methods in the many subfields, normalizations have been studied extensively. They are used to improve the training characteristics of neural networks, making them faster to train and better at generalizing [Huang et al., 2023]. The same applies to GNNs, where normalization plays a key role [Zhou et al., 2020a, Cai et al., 2021, Chen et al., 2022, Rusch et al., 2023a]. However, selecting the correct reference group to normalize jointly is key. For example, a learnable grouping is employed in Deep Group Normalization (DGN), where normalization is performed within each cluster separately [Zhou et al., 2020b]. The employed soft clustering of DGN is only of limited suitability to fostering distinct representations of the node

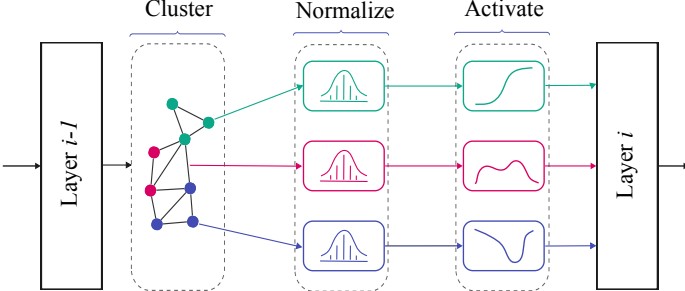

Figure 3: **The components of CNA modules:** They cluster node features without changing the adjacency matrix, normalize them separately, and finally activate with distinct learned functions.

features. Instead, we argue that simple hard clustering, for example, provided by the classic $k$-means algorithm, is sufficient and more desirable. Zhao and Akoglu [2020] suggest PairNorm, where layerwise normalization ensures a constant total pairwise squared distance of node features. Instead of adjusting the node features against collapse, Caso et al. [2023] rewire the topology based on clusters of node features. For the case of multi-graph datasets, Cai et al. [2021] provide a good overview of existing approaches, and argue that normalization shall be performed per graph.

**Learnable Activation Functions.** Using non-polynomial activation functions is crucial for neural networks to be universal function approximators [Leshno et al., 1993]. While most works use rectified-based functions like ReLU, GeLU, SiLU, etc., there are also attempts at learning some limited shape parameters as in PReLU or Swish [Apicella et al., 2021]. There has since been further work on learnable activations with reduced flexibility, namely LEAFs [Bodyanskiy and Kostiuk, 2023], a combination of polynomials and exponentials. However, one can even learn the overall shape of the activations, as demonstrated by rational activation functions [Molina et al., 2019, Boulle et al., 2020, Trimmel et al., 2022]. They have proven to be very helpful in a diverse set of applications, in particular, due to their inherently high degree of plasticity during training [Delfosse et al., 2024]. More importantly, rationals are smoothly differentiable universal function approximators [Molina et al., 2019, Telgarsky, 2017], for which reason we select them as flexible activation functions for CNA. Furthermore, changing the activation function has been found beneficial against oversmooting by Kelesis et al. [2023] too, which increased the slope of the classic ReLU activation to reduce oversmoothing in MPNNs. This further motivates taking a closer look at activations such as done by Khalife and Basu [2024] and in this work.

## 3  Cluster-Normalize-Activate Modules

This section will formally define CNA modules and discuss their design. Adaptive control of the information flow is a promising approach to limit oversmooting in GNNs. We, therefore, propose learning an adaptive node feature update, ensuring distinct node feature representations during the iterative exchange, aggregation, and projection. This benefits the maintenance of effective information propagation in deeper layers. We start by introducing the notation used throughout this work, proceed to recall message-passing GNNs, and finally highlight the three main components of CNA. The overall module is shown in Figure 3.

**Notation.** We consider undirected graphs $\mathcal{G} = (\mathcal{V}, \mathcal{E})$, where the edges $\mathcal{E} \subseteq \mathcal{V} \times \mathcal{V}$ are unordered pairs $\{i, j\}$ of nodes $i, j \in \mathcal{V}$. The set of neighbors of a node $i \in \mathcal{V}$ is denoted as $\mathcal{N}_i = \{j \in \mathcal{V} | \{i, j\} \in \mathcal{E}\} \subseteq \mathcal{V}$. We additionally identify each node $i \in \mathcal{V}$ with a feature vector $\boldsymbol{x}_i \in \mathbb{R}^d$. Together, these form the feature matrix $\boldsymbol{X} \in \mathbb{R}^{d \times |\mathcal{V}|}$, where each column represents the features of a single node. Similarly, depending on whether we model a node-level classification, property prediction, or regression task, we have corresponding target vectors $\boldsymbol{y}_i \in \mathbb{R}^t$, with the special case of $t = 1$ for classification. The target matrix for all nodes is $\boldsymbol{Y} \in \mathbb{R}^{t \times |\mathcal{V}|}$ or a vector for graph-level.

**Message-Passing Neural Networks (MPNNs).** The most prevalent type of GNNs are MPNNs, with GCN, GAT, and GraphSAGE as their best-known representatives. They iteratively transform a graph by a sequence of $L$ layers $\phi = \phi_L \circ \cdots \circ \phi_1$, with $\boldsymbol{Y} = \phi(\mathcal{G}, \boldsymbol{X})$ [Zhou et al., 2020a, Lachaud et al., 2022]. In each layer $\ell$, two steps of computation are performed. First, the node

features $\boldsymbol{h}_j^{(l)}$ of the neighbors $j \in \mathcal{N}_i$ of each node $i \in \mathcal{V}$ are aggregated into a single vector $\hat{\boldsymbol{h}}_i^{(\ell)} = \text{Aggregate}(\{\{\boldsymbol{h}_j^{(l)} \,|\, j \in \mathcal{N}_i\}\})$. Importantly, the Aggregate operation must be invariant to permutations of the neighbors. Popular choices include the point-wise summation or averaging of feature vectors across all neighbors of a node. Second, these features $\hat{\boldsymbol{h}}_i^{(\ell)}$ are projected jointly with the previous node features, as $\boldsymbol{h}_i^{(\ell+1)} = \text{Update}(\boldsymbol{h}_i^{(l)}, \hat{\boldsymbol{h}}_i^{(\ell)})$. The resulting node features $\boldsymbol{h}_i^{(\ell+1)}$ then form the input to the next layer. Both the Aggregate and the Update steps can be learned, where the latter is often instantiated by Multi-layer Perceptrons (MLPs). Note that the features of the very first layer are simply the node features $\boldsymbol{h}^{(1)} = \boldsymbol{X}$, and the resulting last hidden representation is our target output: $\boldsymbol{h}^{(L)} = \boldsymbol{Y}$.

We propose improving the Update-step to elevate the effectiveness of the overall architecture. Usually, the learned projection ends with a non-linear activation, like ReLU. Instead, we propose performing the three steps of CNA, which we will outline below. We want to emphasize that our general recipe is applicable to any MPNN following the above structure.

## 3.1 Step 1: Cluster

The node features of typical graph datasets can be clustered into groups of similar properties. In the case of classification problems, a reasonable clustering would at least partially recover class membership. Note that this unsupervised procedure does not require labels and is applicable to a wide range of tasks. So, even in regression tasks, the target output for each node will usually differ; therefore, partitioning nodes into groups of similar patterns is advantageous, too. We, therefore, cluster the nodes by their features $\boldsymbol{x}_i$ to obtain $K$ groups $\mathcal{C}_1, \ldots, \mathcal{C}_K$ at the end of each Update-step. This separation allows us to then normalize representations and learn activation functions that are specific to the characteristics of these subsets of nodes. It is important to note that the geometry, i.e., the arrangement of edges between nodes, does not change in the progression through GNN layers, while the features associated with each node do. Likewise, cluster membership does not necessarily indicate node adjacency and thus allows learning on heterophilic data as well. Note that this approach is, therefore, distinct from the graph partitioning often performed to shard processing of graphs based on its geometry [Chiang et al., 2019].

In principle, any clustering algorithm yielding a fixed number of clusters $K$ can be used to group the node features. Popular choices include the classic $k$-means [MacQueen, 1967] and Gaussian Mixture Model (GMM) algorithms [Bishop, 2006], which estimate spherical and elliptical clusters, respectively. However, we need to pay attention to the computational costs of such operations. Typical definitions of $k$-means run in $\mathcal{O}(|\mathcal{V}|Kd)$ per iteration [Manning et al., 2009]. Expectation-maximization can be used to learn GMM clusters in $\mathcal{O}(|\mathcal{V}|Kd^2)$ per iteration [Moore, 1998]. We found that the more expensive execution of GMMs did not materialize in substantial improvements in downstream tasks. We, therefore, opted to use a fast implementation of $k$-means. This confirms that k-means often provides decent clustering in practical settings and is sufficiently stable [Ben-David et al., 2007]. In our work, we compared nodes by their Euclidean distance, which we found to work reliably in our experiments. However, CNA permits the flexible use of different and even domain-specific data distances.

## 3.2 Step 2: Normalize

To ensure even scaling of the data across layers, we perform normalization per cluster $\mathcal{C}_k$ and per feature $j$ across all nodes $i \in \mathcal{C}_k$ separately:

$$\widetilde{x}_{ij} = \frac{x_{ij} - \mu_{kj}}{\sqrt{\sigma_{kj}^2 + \epsilon}}, \quad \mu_{kj} = \frac{1}{|\mathcal{C}_k|} \sum_{p \in \mathcal{C}_k} x_{pj}, \quad \sigma_{kj}^2 = \frac{1}{|\mathcal{C}_k|} \sum_{p \in \mathcal{C}_k} \left( x_{pj} - \mu_{kj} \right)^2, \tag{1}$$

where $\epsilon$ is introduced for numerical stability. We want to emphasize that this step is similar to Instance Normalization, yet is nonparametric and does not apply the usual affine transformation to restore the unique cluster representation [Huang et al., 2023]. Similarly, it is not required to scale the mean we subtract as in GraphNorm [Cai et al., 2021]. Instead, we learn a much more powerful transformation in the subsequent Activate step, which subsumes the expressivity of a normal affine projection and thus renders it redundant. The idea of normalizing per cluster $\mathcal{C}_k$ is related to GraphNorm. However, instead of normalizing per graph in the batch, we propose normalizing per cluster within each graph, yet with the same motivation of maintaining the expressivity of the individual node features.

### 3.3 Step 3: Activate

Using an element-wise non-polynomial activation function is crucial for MLPs to be universal function approximators [Leshno et al., 1993]. To maintain distinct representations of node features at large depths, we employ learnable activation functions. Specifically, we use rational activations [Molina et al., 2019] of degree $(m, n)$:

$$R(x) = \frac{P(x)}{Q(x)} = \frac{\sum_{k=0}^{m} a_k x^k}{1 + |\sum_{k=1}^{n} b_k x^k|}. \tag{2}$$

Their purpose is twofold: Firstly, they act as non-polynomial element-wise projections to increase the representational power of the model. Secondly, they replace and subsume the affine transformation in the typical Instance Normalization formulation. Additionally, their strong adaptability allows for appropriate learnable adjustments in the dynamic learning of deep neural networks. This is in line with the findings of Kelesis et al. [2023], who increased the slope of ReLU activations to combat overfitting. Our rationals subsume their approach by further lifting restrictions on the activation function and tuning the slopes automatically while learning the network.

Removing activation functions from GNN layers altogether can–surprisingly–improve overall performance due to reduced oversmoothing [Wu et al., 2019]. Our CNA modules limit oversmoothing further, maintaining strong representational power even in deeper networks. We will demonstrate this in the next section.

### 3.4 Theoretical Underpinnings

We first show how previous proofs of the necessary occurrence of oversmoothing in vanilla GNNs are not applicable when CNA is used. Next, we explain why these proofs are not easily reinstated by illustrating how CNA breaks free of the oversmoothing curse.

**Previous Theoretical Frameworks**  The Rational activations of CNA trivially break the assumptions of many formalisms due to their potential unboundedness and not being Lipschitz continuous. This includes Prop. 3.1 of Rusch et al. [2023b], where, however, the core proofs on oversmoothing are deferred to Rusch et al. [2022]. Again, the activation $\sigma$ is assumed to be point-wise and further narrowed to ReLU in the proof in Appendix C.3. Regarding the more recent work of Nguyen et al. [2023], we again note that CNA violates the assumptions neatly discussed in Appendix A. The CNA module can either be modeled as part of the message function $\psi_k$ or as part of the aggregation $\oplus$. However, in both cases, the proof of Prop. 4.3 (which is restricted to regular graphs) breaks down. In the former case, there appears to be no immediate way to repair the proof of Eq. (15) in Appendix C.3. In the latter case, providing upper bounds in Appendix C.2 is much more difficult.

**How CNA Escapes Oversmoothing**  Restoring the proofs for the occurence of oversmoothing is difficult because CNA was built precisely to break free of the current limitations of GNNs. This can be seen by considering two possible extremes that arise as special cases of CNA. Consider a graph with $N$ nodes. On one end of the spectrum, we can consider CNA with $K = N$ clusters and Rationals that approximate some common, fixed activation, such as ReLU. This renders the normalization step ineffective and exactly recovers the standard MPNN architecture, which is known to be doomed to oversmooth under reasonable assumptions [Rusch et al., 2022, Nguyen et al., 2023]. The same holds with only a single cluster ($K = 1$), i.e., MPNNs with global normalization [Zhou et al., 2020b]. Conversely, we can consider $K = N$ clusters, but now with fixed distinct Rational activations given by $R_i(x) = i$ for each cluster $i \in 1, \ldots, N$. The Dirichlet energy of that output is constant, lower-bounded, and, therefore, does not vanish, no matter the number of layers. In practice, we employ, of course, between $K = 1$ one and $K = N$ clusters and thereby trade off the degree to which the GNN is affected by oversmoothing. The following section will investigate this and other questions empirically.

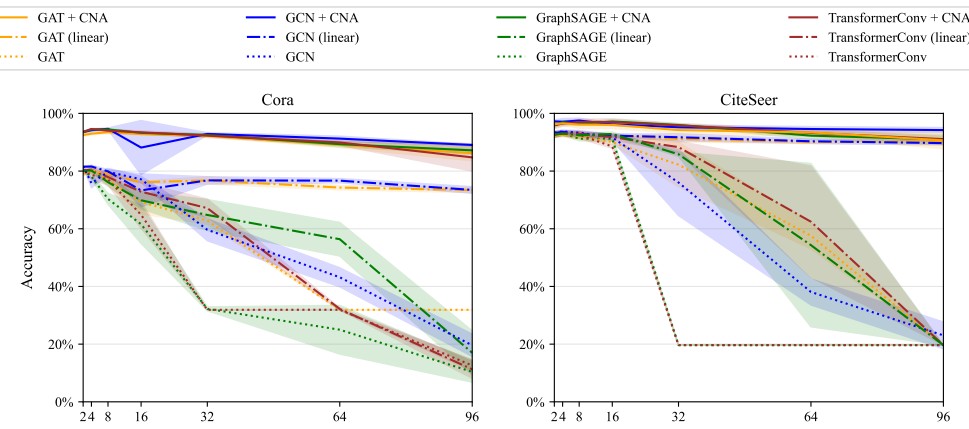

Figure 4: **CNA limits oversmoothing and improves the performance of deep GNNs.**

# 4 Experiments

To evaluate the effectiveness of CNA with GNNs, we aim to answer the following research questions:

(Q1) Does CNA limit oversmoothing?

(Q2) Does CNA improve the performance in node classification, node regression, and graph classification tasks?

(Q3) Can CNA allow for having fewer parameters while maintaining strong performance when scaling to very large graphs?

(Q4) Model Analysis: How important are each of the three steps in CNA? How do hyperparameters affect the results?

**Setup.** We implemented CNA based on PyTorch Geometric [Fey and Lenssen, 2019] to answer the above questions. We searched for suitable architectures among Graph Convolutional Network (GCN) [Kipf and Welling, 2016], Graph Attention Network (GAT) [Veličković et al., 2018], Sample and Aggregate (GraphSAGE) [Hamilton et al., 2017], Transformer Convolution (Transformer-Conv) [Shi et al., 2021b] and Directional GCN (Dir-GNN) [Rossi et al., 2023]. They offer diverse approaches to information aggregation and propagation within graph data, catering to a wide range of application domains and addressing specific challenges inherent to graph-based tasks. Details on the choice of hyperparameters and training settings are provided in Appendix A.2. Average performances and standard deviations are over 5 seeds used for model initialization for all results, except for Tables 1 and 6, where we used 20.

**(Q1) Limiting Oversmoothing.** Since the phenomenon occurs only within deep GNNs, we systematically increased the number of layers in node classification. We mainly compare vanilla GNNs with ReLU to GNNs with CNA. To complete the analysis, we also consider linearized GNNs without any activation function, since they were found to be more resilient against oversmoothing at the expense of slightly reduced performance [Wu et al., 2019]. Figure 4 shows the resulting accuracies for depths of 2 to 96. We can confirm the strong deterioration of vanilla GNNs at greater depths and the partial resilience of linearized GNNs. On the other hand, CNA modules limit oversmoothing drastically and are even more effective than linearized models. At the same time, they significantly alleviate the model's performance shortcomings, effectively eliminating the practical relevance of oversmoothing.

**(Q2) Node Classification, Node Regression, and Graph Classification.** We evaluated CNA by incorporating it into existing architectures and compared the resulting performances with the unmodified variants. As the results in Table 1 demonstrate, our CNA modules significantly improve classification performance on the Cora dataset [McCallum et al., 2000] by up to 13.53 percentage points. Moreover, this improvement shows across different architectures, highlighting CNA's versatility. Next, we extend our analysis to many more datasets and compare CNA to the best-known models from the literature. Specifically, we evaluate the performance on the following datasets: Cora,

Table 1: **CNA consistently increases the accuracy** (↑) of each architecture on Cora.

| Architecture | Baseline | CNA |
|---|---|---|
| GCN | 81.59±0.43 | **93.66±0.48** |
| GraphSAGE | 80.58±0.49 | **93.68±0.50** |
| TransformerConv | 79.97±0.78 | **93.50±0.58** |
| GAT | 80.57±0.81 | **92.94±0.71** |

Table 2: **CNA systematically improves graph classification accuracy** (↑).

| Graph Dataset | Baseline | CNA |
|---|---|---|
| Mutag | 78.42±6.55 | **81.60±4.18** |
| Enzymes | 36.97±3.08 | **50.00±3.25** |
| Proteins | 72.72±2.60 | **74.44±2.49** |

Table 3: **CNA reduces the NMSE** (↓) on two multiscale node regression datasets.

| Model | Chameleon | Squirrel |
|---|---|---|
| GCN | 0.207±0.039 | 0.143±0.039 |
| GAT | 0.207±0.038 | 0.143±0.039 |
| PairNorm | 0.207±0.038 | 0.140±0.040 |
| GCNII | 0.170±0.034 | 0.093±0.031 |
| $G^2$-GCN | 0.137±0.033 | 0.070±0.028 |
| $G^2$-GAT | 0.136±0.029 | 0.069±0.029 |
| Trans.Conv | 0.133±0.033 | 0.072±0.025 |
| **Trans.Conv+CNA** | **0.131±0.033** | **0.068±0.027** |

Table 4: **Comparison of our method CNA with the leaderboard on Papers with Code (PwC),**[2] as of writing on a diverse set of node classification datasets from five typical collections. CNA outperforms the respective leaders, and thereby all compared methods, in eight out of eleven cases (73%). For some, it does so by a significant margin, e.g., on the popular *Cora* and *CiteSeer* datasets.

| | Best CNA Result | | PwC Leaderboard | |
|---|---|---|---|---|
| **Dataset** | **Architecture** | **Accuracy (↑)** | **Architecture** | **Accuracy (↑)** |
| Chameleon | Dir-GNN | **85.86±1.80** | DJ-GNN [Begga et al., 2023] | 80.48±1.46 |
| CiteSeer | GAT | **95.75±0.58** | ACMII-Snowball-2 [Luan et al., 2022] | 82.07±1.04 |
| Computers | TransformerConv | **92.68±0.27** | Exphormer [Shirzad et al., 2023] | 91.47±0.17 |
| Cora | GraphSAGE | **94.18±0.33** | SSP [Izadi et al., 2020] | 90.16±0.59 |
| CoraFull | TransformerConv | **71.82±0.25** | CoLinkDist [Luo et al., 2021] | 70.32 |
| DBLP | GCN | **86.90±0.45** | GRACE [Zhu et al., 2020] | 84.2±0.1 |
| Photo | TransformerConv | **95.96±0.29** | CGT [Hoang and Lee, 2023] | 95.73±0.84 |
| Pubmed | TransformerConv | 90.16±0.13 | ACM-Snowball-3 [Luan et al., 2022] | **91.44±0.59** |
| Squirrel | Dir-GNN | **77.47±1.28** | Dir-GNN [Rossi et al., 2023] | 75.31±1.92 |
| Texas | GraphSAGE | 90.00±3.65 | 2-HiGCN [Huang et al., 2024] | **92.45±0.73** |
| Wisconsin | TransformerConv | 89.29±2.26 | 5-HiGCN [Huang et al., 2024] | **94.99±0.65** |
| #Wins | | **8**/11 | | 3/11 |

CoraFull [Kipf and Welling, 2016], CiteSeer [Bojchevski and Günnemann, 2018], PubMed [Sen et al., 2008], DBLP [Tang et al., 2008], Computers and Photo [Shchur et al., 2019], Chameleon, Squirrel, Texas, and Wisconsin [Pei et al., 2020]. The results in Table 4 demonstrate the effectiveness of CNA. Out of 11 of those datasets, CNA outperforms the SOTA on 8 of them. In particular, for CiteSeer, CNA achieves a classification accuracy of 95.75% compared to 82.07% for ACMII-Snowball-2. This suggests that CNA is particularly effective in dealing with the imbalanced class distribution in CiteSeer. The application of CNA is successful on the famous Cora dataset, achieving 94.18% accuracy compared to the 90.16% of SSP. Considering the results in relation to the dataset properties listed in Appendix A.1, we can see that CNA is particularly effective on larger datasets and such ones with many features. It is largely unaffected by the usually detrimental degree of heterophily and the number of classes due to the clustering step being mostly independent of them.

Table 3 displays the comparison in performance in multi-scale node regression task as considered by Rusch et al. [2023b] on the Chameleon and Squirrel datasets [Rozemberczki et al., 2021]. Here, *multi-scale* refers to the wide range of regression targets from $10^{-5}$ to 1. CNA modules consistently outperform alternative methods in terms of normalized mean squared error (NMSE) based upon the ten pre-defined splits by Pei et al. [2020]. This superior performance highlights the effectiveness of

---

[2] https://paperswithcode.com/task/node-classification

Table 5 & Figure 5: **CNA allows for (i) compact and (ii) accurate models:** The separate treatment of super-nodes boosts expressivity, making GNNs more compact.

| Model | Accuracy (↑) | Params (↓) |
|---|---|---|
| GraphSAGE[*] | 59.97±0.33 | 34.8k |
| GCN[*] | 69.66±0.27 | 388k |
| GraphSAGE | 71.49±0.27 | 219k |
| GCN | 71.74±0.29 | 143k |
| **(i) GraphSAGE+CNA** | **71.79±0.08** | **34.9k** |
| DAGNN | 72.09±0.25 | 43.9k |
| DeeperGCN | 72.32±0.27 | 491k |
| GCNII | 72.74±0.16 | 2.15M |
| RevGCN-Deep | 73.01±0.31 | 262k |
| GAT | 73.91±0.12 | 1.44M |
| UniMP | 73.97±0.15 | 687k |
| RevGAT-Wide | 74.05±0.11 | 3.88M |
| RevGAT-SelfKD | 74.26±0.17 | 2.10M |
| **(ii) GCN+CNA** | **74.64±0.13** | **389.2k** |

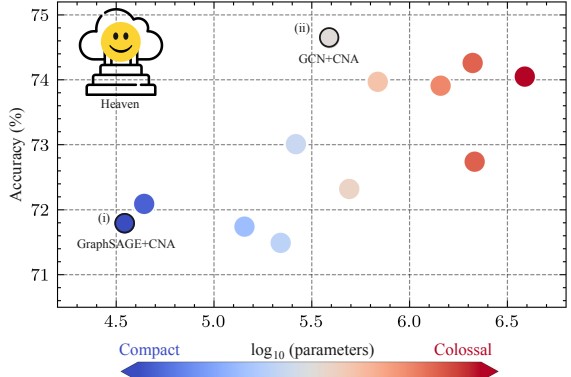

[*] Reproduced by ourselves with the same hyperparameters as (i) and (ii). Other baselines are taken from the literature.

our approach in handling the complexities of node-level regression tasks. These results suggest that our approach has the potential to provide more accurate predictions in real-world scenarios.

To go beyond node-wise tasks on single graphs, we continue by evaluating CNA on graph-level classification tasks. Namely, we compared CNA with ReLU on the Mutag [Debnath et al., 1991], Enzymes [Borgwardt et al., 2005], and Proteins [Borgwardt et al., 2005] datasets from the TUDataset collection Morris et al. [2020]. Table 2 demonstrates that CNA boosts performance unanimously; for instance, achieving an impressive improvement of 13 percentage points on Enzymes.

A further comparison of CNA to other graph normalization techniques is provided in Appendix A.3. Summarizing the findings on node classification, node regression, and graph classification benchmarks, we can confidently answer (Q2) affirmatively.

**(Q3) Parameter Parsimony.** CNA creates super-nodes in graphs, each rescaled separately and governed by an individual learnable activation function. This increased specificity and, in turn, expressivity might allow for more compact models. To investigate this, we use the ogbn-arxiv dataset from the *Open Graph Benchmark* (OGB) [Hu et al., 2020] and follow the setup of Li et al. [2021]. We compare GNNs equipped with CNA to a set of baselines without it. The results in Table 5, first of all, clearly show how CNA outperforms a range of existing GNN models (ii). It achieves a test accuracy of 74.64% while estimating a modest number of learnable parameters (389.2k). This indicates that CNA can successfully capture the underlying patterns in the graph data while maintaining a computationally efficient model. The baselines have varying levels of complexity, with some having more layers and/or channels per layer than others. However, CNA outperforms all competitors, even those with more complex architectures. Figure 5 shows that architectures coming close to the performance of CNA need far more parameters that require learning by gradient descent. Namely, improving GraphSAGE + CNA by 2.47 percentage points (the difference to RevGAT-SelfKD) results in a model about 60x bigger. Similarly, the 2.85 percentage point improvement from GraphSAGE + CNA to GCN + CNA is achieved with a model only about eleven times larger. Additional data, such as the underlying abstract texts originally used to generate the citation graph node features, has recently been used with LLMs to distill additional context [He et al., 2023, Duan et al., 2023]. We exclude them to maintain a level playing field, yet recognize it as an interesting avenue for future work. We argue that CNA modules pave the way for a desirable development of GNN modeling when increasing expressivity would not require an explosion in the number of learnable parameters.

**(Q4) Model Analysis.** We assess the contribution of each of the three operations – Cluster, Normalize, and Activate. To this end, we tested GCN with different subsets of the three operations on the Cora dataset. Table 6 demonstrates that dropping even one of the operations results in minor or no improvement over the plain architecture using ReLU as activation. Cluster-Normalize already improves over the baseline, confirming the findings of Zhou et al. [2020b]. To assess the sensitivity of CNA to the choice of its hyperparameters, we compared the effect of the number of hidden features and the number of clusters per layer on the Cora dataset using GCN, as shown in Figure 6. We

Table 6: **Ablation Study** measured in accuracy (↑) on two datasets.

| Cluster | Normalize | Activate | Cora | obgn-arxiv |
|---|---|---|---|---|
| | | | 81.59±0.43 | 69.65±0.19 |
| ✓ | | | 81.25±0.64 | 69.65±0.19 |
| ✓ | ✓ | | 93.02±0.36 | 69.47±0.36 |
| ✓ | | ✓ | 81.64±0.61 | 69.42±0.15 |
| | | ✓ | 81.49±0.54 | 69.36±0.13 |
| | ✓ | | 81.60±0.72 | 69.66±0.21 |
| | ✓ | ✓ | 81.60±0.70 | 69.42±0.13 |
| ✓ | ✓ | ✓ | **93.66±0.48** | **74.16±0.33** |

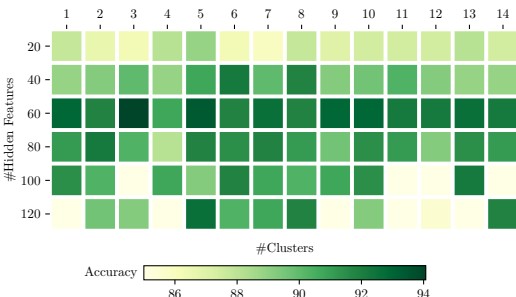

Figure 6: **Hyperparameter sensitivity analysis.**

find that CNA is very robust to the choice of these hyperparameters and works best with moderate numbers of features, as the results from (Q3) would suggest. Answering (Q4), we observed that all three operations of CNA are necessary for the method's efficacy, and it permits practitioners to choose hyperparameters flexibly.

## 5 Conclusions

In this work, we proposed Cluster-Normalize-Activate modules as a drop-in method to improve the Update step in GNN training. The experimental results demonstrated the effectiveness of CNA modules in various classification, node-property prediction, and regression tasks. Furthermore, we found it to be beneficial across many different GNN architectures. CNA permits more compact models on similar or higher performance levels. Although CNA does not entirely prevent oversmoothing, it does considerably limit its effects in deeper GNNs. Our ablation studies have shown that each step in CNA contributes to the overall efficacy and its overall robustness. CNA provides a simple yet effective way to improve the performance of GNNs, enabling their use in more challenging applications, such as traffic volume prediction, energy grid modeling, and drug design.

**Limitations.** We focused our evaluation on very popular architectures and datasets. While it is likely that CNA is beneficial in many other configurations, we did not evaluate its effects on GNNs that are not convolutional MPNNs. Similarly, while we did scale or method to the ogbn-arxiv dataset with about 169k nodes and more than a million edges, yet larger datasets might require further work on the speed of the clustering procedure. Our experiments suggest that oversmoothing is of limited practical relevance. Yet, we did not scale this investigation to even greater depth or establish a formal link to existing theories for oversmoothing.

**Future Work.** The presented results motivate further enhancing CNA in multiple ways. Notably, there are three possible directions. Firstly, regarding clustering, we investigated $k$-means and GMMs, yet it is important to consider other algorithms. For example, Differentiable Group Normalization [Zhou et al., 2020b] is a promising direction for introducing a learnable clustering step. Further, clustering algorithms need not only to yield a fixed number of clusters $k$, but should also produce equally sized clusters. Beyond discovering more stable super nodes, this is likely to improve the learning of the rational projections as well. Apart from representational power, investigating faster clustering procedures paves the way toward scaling GNNs via CNA to dynamic and continuous training settings. Secondly, even more potential for improvement lies in combining CNA with other techniques. For example, representing the Aggregate step as learnable sequence models [Hamilton et al., 2017]. These can be beneficial to distill local information to a greater degree, which in turn could further improve performance and limit oversmoothing. Also, combining CNA with established methods like Edge Dropout or Global Pooling can yield compounding benefits. Finally, the abstract idea behind CNA, namely grouping representations and performing distinct updates, is a more general concept and applicable beyond the architectures we have considered in this work. For instance, Transformers [Vaswani et al., 2017] are known to be equivalent to MPNNs on fully connected graphs and can similarly exhibit oversmoothing [Shi et al., 2021a], motivating a closer look at this connection. Unifying the theory about the different clustering-based normalization approaches and their effect on expressivity and phenomena such as oversmoothing might uncover further opportunities for improvements.

## Acknowledgments and Disclosure of Funding

This research project was funded by the ACATIS Investment KVG mbH project "Temporal Machine Learning for Long-Term Value Investing" and the German Federal Ministry of Education and Research (BMBF) project KompAKI within the "The Future of Value Creation – Research on Production, Services and Work" program (funding number 02L19C150) managed by the Project Management Agency Karlsruhe (PTKA). The Eindhoven University of Technology authors received support from their Department of Mathematics and Computer Science and the Eindhoven Artificial Intelligence Systems Institute. Authors thank Ponturo Consulting AG for their support.

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

# A  Appendix

## A.1  Details on the datasets

An overview of the datasets used in our evaluation is found in Table 7 and in Table 8. In addition to the number of nodes, edges, features, and classes, we also provided the node homophily ratio and whether the classes are distributed uniformly, as well as number of graphs for graph-level datasets. The node homophily ratio measures how many of a node's neighbors are members of the same class. It is computed as:

$$\frac{1}{|\mathcal{V}|} \sum_{i \in \mathcal{V}} \frac{|\{\{i, j\} \in \mathcal{E} \mid j \in \mathcal{N}_i \wedge y_i = y_j\}|}{|\mathcal{N}_i|}.$$

Table 7: **The datasets we used to evaluate CNA.** The table contains statistics for the node classification and property prediction tasks. For regression, we used the Chameleon and Squirrel datasets, but with each page's log average web traffic as the target value.

| Name | #Nodes | #Edges | #Features | #Classes | Homophily | Balanced |
|------|--------|--------|-----------|----------|-----------|----------|
| Chameleon | 2,277 | 36,101 | 2,325 | 5 | 0.10 | No |
| CiteSeer | 3,327 | 9,104 | 3,703 | 6 | 0.71 | No |
| Computers | 13,752 | 491,722 | 767 | 10 | 0.79 | No |
| Cora | 2,708 | 10,556 | 1,433 | 7 | 0.83 | No |
| CoraFull | 19,793 | 126,842 | 8,710 | 70 | 0.59 | No |
| DBLP | 17,716 | 105,734 | 1,639 | 4 | 0.81 | No |
| Photo | 7,650 | 238,162 | 745 | 8 | 0.84 | No |
| PubMed | 19,717 | 88,648 | 500 | 3 | 0.79 | No |
| Squirrel | 5,201 | 217,073 | 2,089 | 5 | 0.09 | Yes |
| Texas | 183 | 309 | 1,703 | 5 | 0.07 | No |
| Wisconsin | 251 | 499 | 1,703 | 5 | 0.17 | No |
| ogbn-arxiv | 169,343 | 1,166,243 | 128 | 40 | 0.43 | No |

Table 8: **The graph-level datasets we used to evaluate CNA.** The table contains statistics for the graph-level classification.

| Name | #Graphs | Avg. #Nodes | Avg. #Edges | #Features | #Classes | Balanced |
|------|---------|-------------|-------------|-----------|----------|----------|
| Mutag | 188 | ~17.9 | ~39.6 | 7 | 2 | No |
| Enzymes | 600 | ~32.6 | ~124.3 | 3 | 6 | Yes |
| Proteins | 1,113 | ~39.1 | ~145.6 | 3 | 2 | No |

## A.2  Details on hyperparameters and training

Throughout the implementation, we used the following software packages:

- PyTorch Geometric (PyG) [Fey and Lenssen, 2019] [3] is widely used for machine learning on Graphs and was our first choice.
- To implement the clustering step, we used Fast PyTorch Kmeans[4], a GPU-based implementation of the renowned algorithm.
- To implement the activate step, we used rationals from the Activation Functions library[5].

The degrees of freedom for the rationals were set to $n = 5$ for the numerator and $m = 4$ for the denominator for all settings. For all experiments, we used the Adam optimizer with weight decay,

---

[3] https://www.pyg.org/
[4] https://github.com/DeMoriarty/fast_pytorch_kmeans
[5] https://github.com/k4ntz/activation-functions

Table 9: **Hyperparameters** for node classification, node property prediction and graph-level classification.

| Dataset | Architecture Type | #Epochs | #Layers | #Clusters | #Hidden | LR | LR Act. | Weight Decay |
|---|---|---|---|---|---|---|---|---|
| Chameleon | Dir-GNN | 500 | 2 | 10 | 500 | $10^{-3}$ | $10^{-8}$ | $5 \cdot 10^{-12}$ |
| CiteSeer | GAT | 100 | 4 | 12 | 60 | $10^{-3}$ | $10^{-5}$ | $1 \cdot 10^{-1}$ |
| Computers | TransformerConv | 100 | 2 | 10 | 20 | $10^{-3}$ | $10^{-5}$ | $5 \cdot 10^{-8}$ |
| Cora | GraphSAGE | 50 | 4 | 12 | 28 | $10^{-3}$ | $10^{-5}$ | $5 \cdot 10^{-6}$ |
| CoraFull | TransformerConv | 100 | 2 | 14 | 140 | $10^{-3}$ | $10^{-5}$ | $1 \cdot 10^{-9}$ |
| DBLP | GCN | 100 | 4 | 12 | 60 | $10^{-3}$ | $10^{-5}$ | $5 \cdot 10^{-1}$ |
| Photo | TransformerConv | 100 | 4 | 8 | 80 | $10^{-3}$ | $10^{-5}$ | $5 \cdot 10^{-2}$ |
| PubMed | TransformerConv | 100 | 2 | 12 | 60 | $10^{-3}$ | $10^{-5}$ | $5 \cdot 10^{-2}$ |
| Squirrel | Dir-GNN | 500 | 2 | 10 | 500 | $10^{-3}$ | $10^{-8}$ | $5 \cdot 10^{-12}$ |
| Texas | GraphSAGE | 200 | 2 | 10 | 100 | $10^{-2}$ | $10^{-5}$ | $1 \cdot 10^{-2}$ |
| Wisconsin | TransformerConv | 200 | 2 | 10 | 500 | $10^{-2}$ | $10^{-5}$ | $5 \cdot 10^{-2}$ |
| ogbn-arxiv | GCN | 300 | 4 | 10 | 400 | $10^{-3}$ | $10^{-5}$ | $1 \cdot 10^{-4}$ |
| ogbn-arxiv | GraphSAGE | 1000 | 4 | 10 | 60 | $10^{-3}$ | $10^{-5}$ | $1 \cdot 10^{-2}$ |
| Mutag | GCN | 1000 | 4 | 8 | 128 | $10^{-3}$ | $10^{-3}$ | $1 \cdot 10^{-4}$ |
| Enzymes | GCN | 1000 | 4 | 8 | 128 | $10^{-3}$ | $10^{-3}$ | $1 \cdot 10^{-4}$ |
| Proteins | GCN | 1000 | 4 | 8 | 128 | $10^{-3}$ | $10^{-3}$ | $1 \cdot 10^{-4}$ |

Table 10: **Hyperparameters** for node regression (TransformerConv), the ablation study (*), and Table 1 (*).

| Architecture Type | #Epochs | #Layers | #Clusters | #Hidden | LR | LR Act. | Weight Decay |
|---|---|---|---|---|---|---|---|
| TransformerConv | 300 | 2 | 8 | 64 | $2 \cdot 10^{-3}$ | $1 \cdot 10^{-5}$ | $1 \cdot 10^{-4}$ |
| * | 200 | 4 | 14 | 280 | $1 \cdot 10^{-3}$ | $1 \cdot 10^{-5}$ | $5 \cdot 10^{-6}$ |

where we set $\beta_1 = 0.9$ and $\beta_2 = 0.999$. We used summation as the aggregation function due to its simplicity and widespread use. Table 9 lists all relevant hyperparameters used for node classification, property prediction, and graph-level classification tasks. Table 10 provides the hyperparameters for Table 1, for the node regression task, as well as for the ablation study. For the sensitivity test, the only difference from (*) in Table 10 was the number of layers, which was set to 32. Regardless of the setting, each experiment was performed on one A100 Nvidia GPU and took between five minutes and two hours, depending on the specific configuration.

### A.3 Comparison to other graph normalization techniques

Table 11 shows how CNA compares to other graph normalization methods proposed in the literature. The reference column indicates the origin of the performance indicators, with the remaining results stemming from Table 1. Our method provides the best classification accuracy.

Table 11: **Comparison of CNA to other graph normalization methods** on the Cora dataset.

| Architecture and Normalization | Reference | Accuracy ($\uparrow$) |
|---|---|---|
| GCN | Baseline | 81.59±0.43 |
| GCN + BatchNorm | Ioffe and Szegedy [2015] | 73.9 |
| GCN + Diff. Group Norm. | Zhou et al. [2020b] | 82.0 |
| GCN + PairNorm | Zhao and Akoglu [2020] | 71.0 |
| GCN + CNA | Ours | **93.66±0.48** |

