# OpenReview forum: "Graph Neural Networks Need Cluster-Normalize-Activate Modules"
_NeurIPS.cc/2024/Conference — NeurIPS 2024 poster_

### Official Review · Reviewer_v2xE · 2024-07-05

**Soundness:** 3
**Presentation:** 2
**Contribution:** 2
**Rating:** 5
**Confidence:** 3

**Summary:**

The paper  introduces a novel plug-and-play module named Cluster → Normalize → Activate (CNA) to enhance the performance of Graph Neural Networks (GNNs). The CNA module is designed to address the issue of oversmoothing, which occurs in deep GNN architectures and limits their ability to solve complex tasks. The module operates by clustering nodes into super nodes, normalizing them, and applying individual activation functions. The authors demonstrate the effectiveness of CNA through extensive experiments on node classification, property prediction, and regression tasks, showing significant improvements in accuracy and a reduction in mean squared error compared to existing methods.

**Strengths:**

* The paper presents a creative solution to a well-known problem in GNNs, oversmoothing, by introducing the CNA module. This approach is a significant advancement in the field of deep learning on graph-structured data.

* The authors provide a thorough empirical evaluation of the CNA module across various tasks and datasets, which substantiates the effectiveness of their proposed method.

**Weaknesses:**

* The paper lacks a theoretical analysis to support the empirical findings. A more rigorous theoretical underpinning could strengthen the claims made about the CNA module's effectiveness. Could the authors provide a theoretical analysis or proof that supports the empirical results?

* The paper does not fully address the computational complexity added by the clustering step in the CNA module in theory or practice, which could be a concern for very large graphs or real-time applications, even with small number of parameters. Are there any optimizations or alternative clustering methods considered to address potential scalability issues?
* In Table 4 why SageConv+CNA has less parameters than GraphSage But  GCNConv+CNA has more parameters than GCN? And the number varies a lot.

**Questions:**

See weaknesses.

**Limitations:**

yes

---

> ### Author Rebuttal · Authors · 2024-08-07
>
> We thank the reviewer for the thoughtful comments and finding our method creative, a significant advancement, and empirical evaluation thorough. We address your concerns next.
>
> ### Q1 (Theoretical underpinnings):
> We agree that an improved understanding of the mechanisms driving CNAs strong empirical performance would be very helpful. Therefore, we added a discussion of the theoretical properties of CNA and its relation to oversmoothing to the global comment above.
>
> In summary, we first show how existing proofs of the necessity of oversmoothing don’t apply to GNNs with CNA. We continue by arguing why they cannot trivially be repaired, since one can construct a hypothetical variant of CNA that certainly inhibits oversmoothing entirely. In practical settings, CNA takes a middle ground between classic oversmoothing GNNs and this potentially immune construction.
> ### Q2 (Scalability):
> We totally agree that scalability is an issue easily overlooked in GNNs. There are two possible approaches to this: (1) Asymptotic behavior and (2) practical scalability.
>
> (1): The runtime of k-means scales linearly with the size of the dataset, which here is the number of nodes in the graphs (cf. lines 187-191 in the submission). Its space requirement is favorable too. This is one reason we chose it over other methods, such as hierarchical clustering, since those methods typically require at least quadratic runtime. We did initially experiment with using Gaussian Mixture Models (which still do scale linearly in the dataset size), yet found their increased overall GNN performance to be insufficient to justify their high practical costs. This leads us to the next reason, its practical performance.
>
> (2): Interested in investigating the practical scaling properties of CNA, we applied it to the rather large *ogbn-arxiv* dataset (e.g., see Table 4). With 169k nodes and 1M edges, it shows that CNA can be applied to large-scale data.
>
> We do recognize the pursuit of scaling to yet larger datasets as one key task for future work. Since the heuristic clusterings of k-means were already sufficient, we can be optimistic about significant further performance improvements. As perfect clustering does not seem to be necessary, subdividing the nodes into independent partitions in which we perform clustering might be feasible while maintaining good overall performance. This would allow for any clustering algorithm to get parallelized. Moreover, even faster procedures like random projections clustering might suffice [Fern et al., 2003].
>
> ### Q3 (Parameter counts in Table 4):
>
> The sole purpose of the experiment in Table 4 was to show that CNA allows for much smaller models at similar performance. To this end, for row (i), we trained a SAGEConv+CNA model to rival the results of SAGEConv without CNA. For a fair comparison, we also show the exact same configuration, but without CNA, in the table below. Similarly, row (ii) shows that GCNConv+CNA can similarly surpass the state-of-the-art with fewer parameters. We also present that model's performance without CNA:
>
> | Model    | Accuracy (↑) | Number of Parameters (↓) |
> | -------- | ------------ | ------------------------ |
> | SAGEConv | 59.97±0.33%  | 34780                   |
> | GCNConv  | 69.66±0.27%  | 388440                  |
>
> ### References
>
> (1) Fern, Xiaoli Zhang, and Carla E Brodley (2003): Random Projection for High Dimensional Data Clustering: A Cluster Ensemble Approach. In Proceedings of the Twentieth International Conference on Machine Learning (ICML-2003). Washington DC.

---

> > ### Comment · Reviewer_v2xE · 2024-08-11
> >
> > Thanks for the authors' responses and my concerns are resolved. I have raised my score to 5

---

> > > ### Author Response · Authors · 2024-08-12
> > >
> > > Dear Reviewer,
> > >
> > > Thank you for acknowledging that we have resolved all your concerns. If there are any further comments from your side, we will be happy to address them before the rebuttal period ends. If there are none, then we would appreciate it if you could reconsider your rating and support our paper for acceptance.
> > >
> > > Regards,
> > >
> > > Authors

---

### Official Review · Reviewer_swbi · 2024-07-07

**Soundness:** 2
**Presentation:** 3
**Contribution:** 2
**Rating:** 5
**Confidence:** 5

**Summary:**

The authors propose a new normalization scheme they call "CNA." They propose clustering the nodes according to their features and computing the normalization statistics for each cluster.
This is essentially a batch norm with groups tailored for graphs.
This normalization is then augmented with a learnable activation function. The three components above are fused together to a "CNA" block to be used after each message passing layer in an MPGNN.
The authors then perform several experiments on node classification and regression tasks.
The authors examine the effect of CNA on the performance of MPGNNs with numerous layers on the Citeceer and Cora datasets. They further perform benchmark experiments, showcasing impressive improvement on the Citeceer dataset.
Afterwards, some additional ablation studies are conducted, considering different subsets of the CNA components showing how the performance is affected.

**Strengths:**

1. I see why the proposed method is reasonable for helping combat over-smoothing, at least in some scenarios.
After all, enforcing normalization for nodes with similar features seems to be stronger than enforcing normalization for all nodes.
2. The presentation is neat, and the writing is satisfying.

**Weaknesses:**

1. Although the "How" is clear, I am not fully convinced by the "Why". The authors did not provide any theoretical insight into why CNA works.
2. I also find the empirical experiments lacking. I would have expected to see many more experiments demonstrating why CNA helps combat over-smoothing, especially pricing analysis on different types of graphs (e.g., homophilic vs. heterophilic, inductive vs. transductive, etc...)
3. The authors didn't examine how the dataset properties affect the effectiveness of CNA
4. The experiments do not convince why CNA is preferred over other normalization layers.
5. I do find the aggregation part of CNA slightly unrelated to the main purpose of the paper and incremental to the results (as clearly indicated by the ablation studies).
6. Some of the benchmark's improvements seem to be statistically insignificant (the improvement in the mean performance seems small compared to the stds).

**Questions:**

Please see the "Weaknesses" section

**Limitations:**

No potential negative societal impact  is observed

---

> ### Author Rebuttal · Authors · 2024-08-07
>
> We thank the reviewer for the thoughtful comments and for finding the writing satisfying. We address your concerns next.
>
> ### Q1 (Why CNA works):
>
> We now provide theoretical underpinnings of why CNA works. Please have a look at the global response.
>
> ### Q2 (Empirical evaluation):
> We already provided empirical results on
>
> - node classification,
> - multi-scale node regression,
> - oversmoothing,
> - parameter parsimony,
> - ablation experiments, and
> - hyperparameter sensitivity.
>
> We focused on transductive node property prediction, following the works of, for example, Zhao et al. (2020), Zhou et al. (2020), and Rusch et al. (2023). We see no fundamental issues with settings where graphs vary. For instance, CNA works well in graph-level tasks too:
>
> | Dataset  | Accuracy with CNA (↑) | Accuracy with ReLU (↑) |
> | -------- | --------------------- | ---------------------- |
> | MUTAG    | **81.60±4.18%**   | 78.42±6.55%            |
> | ENZYMES  | **50.01±3.25%**   | 36.97±3.08%            |
> | PROTEINS | **74.44±2.49%**   | 72.72±2.60%            |
>
> Here, we ran GCNConv with ReLU and compared it with CNA (@ 10 clusters) on three datasets: MUTAG, ENZYMES, and PROTEINS [Morris et al., 2020]. Results are averaged over five seeds. CNA improves classification accuracy for all data sets, confirming the effectiveness of CNA across various tasks.
>
> We now extended the inspection of the occurrence of oversmoothing by visualizing the Dirichlet energy in shallow to even deeper GNNs, which is well-known to measure (over)smoothing [Rusch et al., 2023]. Please see Figure 1 in the rebuttal PDF. We see how the vanilla GNNs typically start in the undersmoothing regime with low DE, transition to an optimal region with elevated DE, and finally deteriorate due to oversmoothing with low DE again. A high DE is a necessary condition for good model performance, as all CNA models and the linearized GATConv and GCNConv prove. This strongly suggests that CNA inhibits oversmoothing at greater depths and thereby unlocks a large portion of its performance increase. The increased expressivity then allows it to achieve better accuracies than the linearized GNNs, combining the best of both worlds.
>
> ### Q3 (Effect of dataset properties):
>
> We are happy to shine more light on the details of our findings. Table 6 in the appendix can hereby help us to get an overview of the dataset properties and statistics. There are different aspects along which we can evaluate the effectiveness of CNA:
>
> - **Number of nodes and edges**: Since CNA increases the model expressivity and capacity, its performance benefit is stronger on larger datasets. In particular, if we look at the results of Tab. 3, classic methods work better on the rather small Texas and Wisconsin tasks. Furthermore, the original ablation study in Tab. 5 was run on Cora, which is only of medium size. If you look at our response to your (Q5), you can see that the benefits are much larger on the huge ogbn-arxiv.
> - **Degree of Homophily**: Our key ingredient CNA is not directly affected by a graph being homo- or heterophilic. This is because the clustering step and subsequent normalization and activation are invariant to the topology of the graph. Indeed, we see good performance on both homophilic graphs such as Cora, and heterophilic ones such as Chameleon. We had these degrees in our paper in Tab. 6 in the appendix.
> - **Number of features**: CNA appears to be able to better leverage large numbers of features. If again looking at Tab. 3, the difference due to CNA on Computers, Photo, and Pubmed is rather small, where the number of features is only moderate. Results tend to be better when more features are available, possibly due to easier clustering.
> - **Number of classes** (for node classification tasks): CNA can successfully be applied to small and large numbers of classes alike, as is apparent when comparing Cora with 7 and CoraFull with 70 classes (CoraFull) as seen in Tab. 3.
>
> ### Q4 (Comparison to other normalization techniques):
>
> Thank you for the question. We provide the new results. Please have a look at the global response.
>
> ### Q5 (Ablation of activation function):
>
> Most commonly employed normalization techniques are followed by a simple scalar transformation, including the prolific BatchNorm, InstanceNorm, LayerNorm, and GroupNorm (see also Huang et al. (2023), specifically Tab. I). We extend this notion with much more powerful transformations, namely Rational activations, which are themselves universal scalar function approximators. This helps to further increase the expressivity of GNNs.
>
> Please also have a look at our response to (Q2) of reviewer gBSJ.
>
> ### Q6 (Statistical significance):
>
> We kindly disagree and concur with the three other reviewers:
> - “CNA module enhances the expressiveness and performance of GNNs” (Reviewer jRzY)
> - “very strong improvement” (Reviewer gBSJ)
> - “showing significant improvements” (Reviewer v2xE)
>
> Table 1 shows that introducing CNA into existing architectures increases their classification performance by consistently more than an impressive 10%. Table 2 shows that CNA achieves the best overall results on regression tasks, too. In Table 3, we compare CNA to the state-of-the-art on an extensive collection of standard benchmark datasets. We find that CNA is the best architecture overall.
>
> ### References
> - Huang et al. Normalization Techniques in Training DNNs: Methodology, Analysis and Application. TPAMI 2023
> - Morris et al. TUDataset: A Collection of Benchmark Datasets for Learning with Graphs. ICML 2020 Workshop
> - Rusch et al. A Survey on Oversmoothing in Graph Neural Networks. arXiv 2023

---

> > ### Comment · Reviewer_swbi · 2024-08-11
> >
> > Dear Authors,
> >
> > After thoroughly reviewing your response to my concerns and those of other reviewers, I believe that the additional clarifications and experiments you provided help to emphasize the proposed method's contribution and quality. As a result, I have raised my score to 5.

---

> > > ### Author Response · Authors · 2024-08-12
> > >
> > > Thank you for the comment and raising the score. We would be happy to answer any further concerns before the rebuttal period ends. If there are none, then we would appreciate if you can reconsider your rating and support our paper for acceptance.

---

### Official Review · Reviewer_gBSJ · 2024-07-09

**Soundness:** 2
**Presentation:** 3
**Contribution:** 2
**Rating:** 5
**Confidence:** 4

**Summary:**

The paper describes a new updating rule based on a sequence of operation clustering, normalization and a learnable activation to replace the original plain relu-like update message passing and empirically show that such learnable updating function gains large performance improvement on existing benchmark datasets. The author claims the new updating rule alleviates oversmoothing to some extent and has better expressive power from a learnable activation function. Compared with existing graph normalization, the learnable involved for normalization in this paper is postponed to the learnable activation stage to ensure a better expressivity. The clustering used here is a simple Kmeans clustering and the paper suggests that simple kmeans is sufficient to guarantee performance and save computation time.

**Strengths:**

1.The paper shows a clear structure of the new updating rules, the writing is clear and the presentation is easy to understand.
2.The empirical experiments suggest a very strong improvement gain in terms of the baseline models, even compared with existing PWC leaderboard, somewhat indicate the effectiveness of the new updating method.
3.The computation resources needed is suggested to be manageable compared with more complex architectures.
4. The idea is simple and seems to be applicable universally to most MPNN structures.

**Weaknesses:**

1. Although the empirical results show a great performance gain on Cora and Citeseer dataset, this CNA module lacks theoretical analysis on why it works and where the performance gain comes from. It is very unintuitive to consider why normalization over a knn based cluster can show a much better performance, there lacks a comparison between existing graph normalization techniques and the proposed one for a better comparision.
2. From the ablation study in Table 5, it suggests that simply with cluster and normalization gives most of the performance gain on Cora data, the   effect of learnable activation modules seems to be negligible. The author claims the expressive power is preserved in this learnable activation part. So why the performance gain mostly come from the first two part (from the results of Table 5)? This is somewhat contradict to the author's claim.
3. To better reflect the method's ability of alleviating oversmoothing, the author should reflect the layer wise dirichlet energy plot for this method and also consider extend the layer number from 32 to at least 64, as produced from previous papers, like G2-gating.
4. The lack of discussion on the clustering method (using only k-means) leads to concern about the stability of the method. K-means clustering is known to be not stable. It is weird there is no discussion about the sensitivity of the choice of cluster number, metric choice, etc.

**Questions:**

As suggested in the weakness. My most concern is on point 2 and 4.

**Limitations:**

The author suggests the lack of large-scale dataset experiments and limited connection to oversmoothing.

---

> ### Author Rebuttal · Authors · 2024-08-07
>
> We thank the reviewer for the thoughtful comments and for acknowledging our strong results, universality of the method, and clear writing. We will address the concerns next.
>
> ### Q1 (Additional theoretical analysis and comparison to other normalization techniques):
> Please see our general response/comment for more theoretical insights.
>
> Regarding the normalization techniques, there indeed are many methods specifically for GNNs, as also discussed in Section 2 on Related Work. In particular, GraphNorm for graph-level tasks (as opposed to node-level ones) provides partial normalization, where features are normalized across each separate graph [Cai et al., 2021]. PairNorm normalizes features to zero mean and constant total pairwise squared distance between nodes [Zhao and Akoglu, 2020]. However, **considering groupings of the nodes can further improve the effectiveness of normalization. This was also shown by Zhou et al. (2020), where learned soft-clustering was performed instead of nonparametric hard k-means as in our method.** Generally, one wants to not only normalize within clusters but also apply a learned transformation [Ioffe and Szegedy, 2015]. This makes the constituents of the cluster distinct from the other ones.
>
> We further note that we replaced the affine transformation following most normalization techniques with a more expressive Rational activation. Overall, CNA proved to be highly effective. Please see comparisons to other normalization techniques in the global comment.
>
> We further added a discussion of theoretical properties related to oversmoothing to the global comment above. We believe that reducing the impact of oversmoothing significantly contributes to the improved performance of CNA observed in our thorough empirical evaluation. See also our response to your third question (Q3) for deeper empirical insights.
>
> ### Q2 (Ablation study):
> Much like with other architectures, different components contribute to the overall performance to differing degrees, depending on the dataset. For example, if the ablation study is performed on the *ogbn-arxiv* dataset instead of on Cora, the necessity of the learned Activate step is much more visible:
> | Ablated variant | Accuracy (↑)        |
> | --------------- | ------------------- |
> | CNA             | **74.64±0.13%** |
> | CN + ReLU | 69.55±0.42% |
>
> The table shows the results of training a GNN with GCNConv with cluster and normalize steps, but with (CNA) and without rational activations (CN+ReLU). We further note that full CNA improves the convergence speed of GNN learning over just CN, as shown in Figure 2 in the rebuttal PDF.
>
> ### Q3 (Alleviating oversmoothing in deeper GNNs):
> Thank you for this astute comment! Please have a look at top 2 graphs of Figure 1 in the rebuttal pdf. There, we reran and extended the results on Cora and CiteSeer beyond 32 layers to 64 and even 96 layers, again with five seeds. This further confirms that CNA maintains high performance even at great depths. Coming in second are again the models with removed activations (“linear”). Lastly, vanilla GNNs quickly deteriorate in performance at increasing depths.
>
> Moreover, we visualized the final Dirichlet energy (DE) of each of the models in the Fig 1 of rebuttal pdf, which is well-known to measure (over)smoothing [Rusch et al., 2023]. We see how the vanilla GNNs typically start in the undersmoothing regime with low DE, transition to an optimal region with elevated DE, and finally deteriorate due to oversmoothing with low DE again. A high DE is a necessary condition for good model performance, as all CNA models and the linearized GATConv and GCNConv prove. This shows that CNA inhibits oversmoothing at greater depths and thereby unlocks a large portion of its performance increase. The increased expressivity then allows it to achieve better accuracies than the linearized GNNs, combining the best of both worlds.
>
> ### Q4 (Stability of k-means):
> Indeed, we can provide a more in-depth discussion of the properties of k-means to justify its use and good empirical performance.
>
> Since we were also interested in the sensitivity of CNA to the number of clusters, we performed a dedicated study showing that good configurations can easily be found by hyperparameter search. We describe those results in lines 300-304 and Figure 6. Overall, this confirms that k-means often provides decent clustering in practical settings and is sufficiently stable [Ben-David et al., 2007]. We also found that using much more computationally expensive Gaussian Mixture Models did not significantly improve the final performance of the GNNs. In our work, we compared nodes by their Euclidean distance, which we found to work reliably in our experiments. However, other applications might use different or even domain-specific distances. This flexibility is a benefit of CNA, allowing for more task context to be used in modeling.
>
> We will add this discussion of the stability of k-means to the main paper.
>
> References
>
> (1) Ben-David et al. Stability of K-Means Clustering. Learning Theory 2007
>
> (2) Cai et al. GraphNorm: A Principled Approach to Accelerating Graph Neural Network Training. ICML 2021
>
> (3) Ioffe et al. Batch Normalization: Accelerating Deep Network Training by Reducing Internal Covariate Shift. ICML 2015
>
> (4) Rusch et al. A Survey on Oversmoothing in Graph Neural Networks. arXiv 2023
>
> (4) Zhao et al. PairNorm: Tackling Oversmoothing in GNNs. ICLR 2020.
>
> (5) Kaixiong et al. Towards Deeper Graph Neural Networks with Differentiable Group Normalization. NeurIPS 2020

---

> > ### Comment · Reviewer_gBSJ · 2024-08-10
> >
> > Thanks for the detailed explanation and experiment results on the Dirichlet energy. I found most of the part convincing. I have, however, one question related to the theoretical proof. I understand that it is non-trivial to provide a rigorous proof on how CNA directly avoids oversmoothing and your analysis on how it is not suitable for the existing case and the extreme case analysis seems convincing. Yet, I found there lacks two important aspects that should be necessarily be investigated:
> > 1. In the survey paper [1], Rusch mentioned that in addition to the Dirichlet energy, expressive power is equally important to improve the performance of the deep GNN model. Therefore, an theoretical analysis on the expressive power of CNA compared with non-CNA module should be provided.
> > 2. Since KNN has shown sufficient good performance for the CNA clustering part, I wonder what exactly is it clustering, any analysis on that part should provide valuable theoretical insights on why this method works. My current intuition is that the cluster is basically a different forms of rewiring from feature space and the normalization applied serve as a similar functionality of feeding the feature rewiring edge information to the nodes.
> > I think the two points should be addressed more in the theoretical analysis as we are easy to observe the Dirichlet energy is well preserved and it is in fact easy to preserve such energy according to [1].
> > Overall, I think the paper is very valuable if the two points can somehow be addressed.
> >
> >
> >
> > [1] Rusch, T., Bronstein, M.M., & Mishra, S. (2023). A Survey on Oversmoothing in Graph Neural Networks. ArXiv, abs/2303.10993.

---

> ### Author Response · Authors · 2024-08-10
>
> Thank you for taking the time to provide the feedback on our rebuttal. In the following, we want to address both aspects you pointed out:
> ### 1. Expressive power of CNA
> The expressive power of Rationals were studied by both Delfosse et al. (1) and Telgarsky (2).  The author of (2) proven Rationals to be better approximants than polynomials in terms of convergence.  All the while, Delfosse et al. (1), show in their that in actuality Rationals amplify a model with high neural plasticity. More to this point, they provided a proof that Rationals can dynamically make use of a residual connection: a Rational embeds a residual connection ⇔ m>n. See p. 4 of the paper (1) for the theorem and its proof. Throughout our experimental evaluations, we used m=5 and n=4, which answers where the expressive power of CNA modules stems from.
> ### 2. Role of Clustering in CNA
> We agree with your intuition. More to the point, Shi at al. (3) demonstrate in their work that clustering minimizes redundant connectivity and so makes message passing more effective. And, as you are pointing out, normalization facilitates feeding the rewiring edge information to the nodes.
>
> We will add these points to the paper. Thanks again, your inputs have helped to make our manuscript more clear. We hope we have addressed all your concerns and it will be great if you can reconsider your score
>
> ##### **References**
> (1) Delfosse et al., Adaptive Rational Activations to Boost Deep Reinforcement Learning. ICLR 2024.
>
> (2) Matus Telgarsky, Neural Networks and Rational Functions. ICLR 2017.
>
> (3) Shi et al., ClusterGNN: Cluster-Based Coarse-To-Fine Graph Neural Network for Efficient Feature Matching. CVPR 2022.

---

> > ### Author Response · Authors · 2024-08-13
> >
> > Dear Reviewer,
> >
> > Thank you again for the follow up question which we had answered a couple of days ago. We hope we have now convinced you of the valuability of our method. Plese do let us know if there are any further concerns from your side.
> >
> > Regards,
> >
> > Authors

---

> > > ### Comment · Reviewer_gBSJ · 2024-08-13
> > >
> > > Dear authors, I appreciate the response. For the expressive power, I believe it should be related to GNN expressive power instead of expressive power as Neural Networks, which can be how the activations helps exceed the limit of WL-test (for graph-level task especially). For the Shi's work, I checked its IEEE version, I am not sure where it discusses the explanation you mentioned. I don't see any theoretical analysis in its work. Can you elaborate on it? Given these, I think this work still lacks some solid theoretical analysis.

---

> ### Author Response · Authors · 2024-08-14
>
> Thank you for the comment!
>
> ### On Expressive Power of Learnable Activations in Context of GNNs
>
> To the best of our knowledge, our work is the first one using Rationals in the context of GNNs. On the other hand, research on learnable activation functions is limited, and our work contributes to this area by exploring the use of Rationals in GNNs.
>
> Some papers do explore the effects of activation functions on the expressive power of GNNs:
>
> Khalife and Basu (1) study the power of graph neural networks and the role of the activation function shows that changing the activation function can drastically change the power of GNNs.
>
> Yu et al. (2) investigate in their work influences of different activation functions in GNNs and find that the choice of activation function can significantly affect the performance.
>
> Although these studies do not specifically focus on learnable activation functions, they do highlight the importance of activation functions in determining the expressive power of GNNs. Both discuss the impact of activation functions on the expressive power of GNNs in the context of the Weisfeiler-Lehman (WL) test. More to this point, they do not specifically address how activation functions can help exceed the limit of the WL-test.
>
> The first paper (1) proves that GNNs with piecewise polynomial activations cannot distinguish certain non-isomorphic graphs, while those with non-piecewise polynomial activations (like sigmoid, hyperbolic tan) can distinguish them in two iterations.
>
> The second paper (2) investigates the expressive power of Graph Transformers with different activation functions (softmax, tanh, sigmoid) and finds that sigmoid is the most powerful, enabling the Graph Transformer to achieve the same level of expressiveness as the WL test.
>
> Molina et al. (3) show that Rationals can approximate either tanh or sigmoid functions, allowing for end-to-end learning of deep networks without the need to manually select fixed activation functions. CNA builds upon these findings.
>
> Neither (1) and (2) nor other related works explicitly discuss how activation functions can help exceed the limit of the WL-test. The focus is primarily on understanding how different activation functions affect the expressive power of GNNs within the bounds of the WL-test.
>
> ### Role of Clustering and Normalization in CNA
> Some papers do explore the concepts of normalization and rewiring in GNNs, providing empirical evidence:
>
> Chen et al. (4) study on learning graph normalization for GNNs discusses the importance of normalization in improving the performance of GNNs.
>
> Caso et al. (5) propose a novel graph rewiring approach to improve GNNs' performance on graph-related tasks.
>
> Zhou et al. (6) offer a review of methods and applications of GNNs, highlighting the role of normalization and rewiring in enhancing the efficiency and effectiveness of GNNs.
>
> Although these studies do not specifically address how normalization facilitates feeding rewiring edge information to nodes in combination with clustering, they do emphasize the importance of normalization and rewiring in improving the performance of GNNs. As far as we are aware, sadly, there are no dedicated theoretical studies on this particular question yet. We suggest that future research could investigate this topic further, potentially leading to new insights and improvements in GNNs.
>
> As far as Shi et al. are concerned, we agree that there is no theoretical analysis there, but the reduction of redundancy is the whole motivation behind the paper, and the experimental results show that.
>
> ### Conclusion
> In conclusion, our proposed method, CNA, contributes to the current state of research on the expressive power of learnable activations in the context of GNNs. We have addressed the reviewer's concerns and provided an overview of the related works on this topic, in extension the **theoretical analysis of the combined CNA modules in the global comment**. While there are no dedicated theoretical studies on the specific role of clustering and normalization in CNA yet, we suggest that future research investigates this topic further.
>
> ### References
>
> (1) Khalife, S., & Basu, A.. On the power of graph neural networks and the role of the activation function. arXiv preprint arXiv:2307.04661.
>
> (2) Yu et al., Activation Function Matters in Graph Transformers, ICLR 2024
>
> (3) Molina et al., Padé Activation Units: End-to-end Learning of Flexible Activation Functions in Deep Networks. ICLR 2020.
>
> (4) Chen et al., Learning Graph Normalization for Graph Neural Networks, Neurocomputing 2022
>
> (5) Caso et al., Renormalized Graph Neural Networks, preprint on arXiv:2306.00707v1
>
> (6) Zhou et al., Graph neural networks: A review of methods and applications, AI Open 2020

---

### Official Review · Reviewer_jRzY · 2024-07-12

**Soundness:** 3
**Presentation:** 4
**Contribution:** 3
**Rating:** 6
**Confidence:** 4

**Summary:**

This paper proposes a novel module, CNA (Cluster-Normalize-Activate), to address the oversmoothing problem in Graph Neural Networks (GNNs).  The CNA module operates in three steps: clustering node features, normalizing them within clusters, and applying learnable activation functions to each cluster.

**Strengths:**

1.	Experiments and analysis are solid. Results show that the CNA module enhances the expressiveness and performance of GNNs, particularly in node classification and regression tasks.

2.	The CNA module can be applied to various GNN architectures.

3.	Incorporating the CNA module requires fewer or comparable parameters than existing SOTA methods.

**Weaknesses:**

1.	In Table 4, I would encourage to report the performance and parameter amount of SAGEConv and GCNConv baselines as well. It could help readers to have a clearer understanding of how many parameters are introduced by the proposed CNA module.

2.	Can the CNA module improve the performance in graph-level tasks?

**Questions:**

See in Weaknesses.

---

> ### Author Rebuttal · Authors · 2024-08-07
>
> We thank the reviewer for the thoughtful comments and for acknowledging that our experiments and analysis are solid and that our method is easily adaptable. We will address your concerns next.
>
> ### Q1 (Results on SAGEConv and GCNConv):
> Thank you for taking a closer look. We have now run new experiments using the same SAGEConv and GCNConv without CNA. Please find the results in the table below.  As it can be seen, the accuracy drops by a huge margin, whereas the change in the number of parameters is minimal.
>
> | Model    | Accuracy (↑) | Number of Parameters (↓) |
> | -------- | ------------ | ------------------------ |
> | SAGEConv | 59.97±0.33%  | 34780                   |
> | GCNConv  | 69.66±0.27%  | 388440                  |
>
> A Rational activation only introduces 10 learnable parameters, since the activation is performed pointwise on each entry of the node feature vector. This means that for 5 clusters and 20 layers, we add 5\*20\*10 = 1000 new learnable parameters.
>
> ### Q2 (CNA on graph-level tasks):
> We gladly extended our evaluation of CNA to graph-level learning, and report the results below. In particular, we ran GCNConv with ReLU and compared it with CNA (@ 10 clusters) on three datasets: MUTAG, ENZYMES, and PROTEINS [Morris et al., 2020]. Averaged over five seeds, CNA improves classification accuracy for all data sets:
> | Dataset  | Accuracy with CNA (↑) | Accuracy with ReLU (↑) |
> | -------- | --------------------- | ---------------------- |
> | MUTAG    | **81.60±4.18%**   | 78.42±6.55%            |
> | ENZYMES  | **50.01±3.25%**   | 36.97±3.08%            |
> | PROTEINS | **74.44±2.49%**   | 72.72±2.6%             |
>
>
> This confirms the effectiveness of CNA across a variety of tasks.
>
> References
>
> (1) Morris, Christopher, Nils M Kriege, Franka Bause, Kristian Kersting, Petra Mutzel, and Marion Neumann (2020): TUDataset: A Collection of Benchmark Datasets for Learning with Graphs. Graph Representation Learning and Beyond (GRL+), ICML 2020 Workshop.

---

> > ### Comment · Reviewer_jRzY · 2024-08-12
> >
> > Thanks for the additional experiments! I will keep my positive score.

---

### Author Rebuttal · Authors · 2024-08-07

### Global Response/Comment
We want to thank all reviewers for their time and effort in improving this work. We particularly appreciate your acknowledgment of the relevance of the issue, the novelty of the approach, and the extensive and convincing experiments. Your comments and questions helped improve the paper, and we hope to have clarified all of them below.

Since there were multiple requests for theoretical analysis, we provide that one here in the global response. We wrote dedicated answers to all your other questions. Furthermore, please see the attached PDF.

**Theoretical Analysis**

It has been suggested to more formally discuss the relationship of CNA to oversmoothing. There are two directions in which we can tackle this. (1) We can show *how* previous proofs of the necessary occurrence of oversmoothing in vanilla GNNs are not applicable when CNA is used. (2) We can provide a reason for *why* these proofs are not easily repairable and how CNA breaks free of the oversmoothing curse.

**(1)** The Rational activations of CNA trivially break the assumptions of many formalisms due to their potential unboundedness and not being Lipschitz continuous. This includes Prop. 3.1 of Rusch et al. (2023), where, however, the core proofs on oversmoothing are deferred to Rusch et al. (2022). There, again, the activation $\sigma$ is assumed to be point-wise and further narrowed to ReLU the proof in Appendix C.3. Regarding the more recent work of Nguyen et al. (2023), we again note that CNA violates the assumptions neatly discussed in Appendix A. The CNA module can either be modeled as part of the message function $\psi_k$ or even as part of the aggregation $\oplus$. However, in both cases, the proof of Prop. 4.3 (which is restricted to regular graphs) breaks down. In the former case, there appears to be no immediate way to repair the proof of Eq. (15) in Appendix C.3. In the latter case, providing upper bounds in Appendix C.2 is not exactly straightforward.

**(2)** This begs the question of why that is difficult. The answer is that we specifically built CNA to break free of the current limitations of GNNs. Empirically, this indeed hurdled oversmoothing. In theory, this makes sense once we consider two possible extremes that arise as special cases of CNA. Consider a graph with $N$ nodes. On one end, we can consider CNA with $N$ clusters and Rationals that approximate some common activation, such as ReLU. This exactly recovers the standard MPNN architecture, which is known to be doomed to oversmoothing under reasonable assumptions (see above). We can alternatively consider the same setting with only one cluster, i.e., MPNNs with global normalization. It is also known to exhibit oversmoothing empirically [Zhou et al., 2020]. Conversely, we can consider $N$ clusters, with constant Rational activations given by $R_i (x) = i$ for each cluster $i \in \{1, \dots, N\}$. Obviously, the Dirichlet energy of that output is lower bounded and does not vanish no matter the number of layers. In practice, we employ, of course, between one and $N$ clusters, and thereby trade off the degree to which the GNN is affected by oversmoothing. Bounding this relationship goes far beyond the scope of the current work, and is definitely an interesting direction for future work.

**Overall motivation: why CNA works**

- Cluster: We first identify groups of nodes with similar properties. We later learn separate projections for each such group, ensuring distinct feature representations. It is reasonable to assume such a structure exists, especially in classification datasets.
- Normalize: To maintain good stability during the propagation through the layers, we normalize the features within each cluster.
- Activate: Finally, we project the nodes with separate learned activations per cluster. These expressive functions also take up the work of the usual learned transformation after normalization steps.

**Comparison to other graph normalization techniques**

We also provide a comparison of CNA to BatchNorm (5), Deep Group Normalization (4), and PairNorm (6):

| Model and Normalization | Accuracy on Cora (↑) |
| ----------------------- | -------------------- |
| GCNConv                 | 82.2%                |
| GCNConv + BatchNorm     | 73.9%                |
| GCNConv + DGN           | 82.0%                |
| GCNConv + PairNorm      | 71.0%                |
| GCNConv + CNA (ours)    | **93.66±0.48%**  |

Results for the other methods were taken from Zhou et al. (2020). You can find details on how other ablated variants of CNA than the ones shown here fare by consulting Table 5. We also note that Table 2 already compares PairNorm with CNA.

**References**

(1) Rusch, T. Konstantin, Benjamin Paul Chamberlain, Michael W. Mahoney, Michael M. Bronstein, and Siddhartha Mishra (2023): Gradient Gating for Deep Multi-Rate Learning on Graphs. ICLR 2023.

(2) Rusch, T Konstantin, Benjamin P Chamberlain, James Rowbottom, Siddhartha Mishra, and Michael M Bronstein (2022): Graph-Coupled Oscillator Networks. ICML 2022. Baltimore, Maryland, USA.

(3) Nguyen, Khang, Hieu Nong, Vinh Nguyen, Nhat Ho, Stanley Osher, and Tan Nguyen (2023): Revisiting Over-Smoothing and over-Squashing Using Ollivier-Ricci Curvature. ICML 2023, 202:25956–79. Honolulu, Hawaii, USA.

(4) Zhou, Kaixiong, Xiao Huang, Yuening Li, Daochen Zha, Rui Chen, and Xia Hu (2020): Towards Deeper Graph Neural Networks with Differentiable Group Normalization. NIPS ’20. Red Hook, NY, USA.

(5) Ioffe et al. Batch Normalization: Accelerating Deep Network Training by Reducing Internal Covariate Shift. ICML 2015

(6) Zhao et al.  PairNorm: Tackling Oversmoothing in GNNs. ICLR 2020

---

### Decision · Program_Chairs · 2024-09-25

**Decision:**

Accept (poster)

**Comment:**

The paper proposes a novel module for GNNs with the goal of overcoming oversmoothing. The new module is referred to as CNA , where "c" stands for clustering nodes according to their features, "n" stands for normalization applied to the clusters, and "a" stands for learnable activation functions. It is demonstrated empirically that the new model achieves new state-of-the art on Cora and Citeseer.

After rebuttal, in which authors provided a set of additional experiments, all reviewers votes were positive (3x5 and 1x6). The main remaining criticism is the missing theory to explain the success of the approach. Due to the great experimental results I nevertheless vote for acceptance.